# Preparation and Properties of Waterborne Polypyrrole/Cement Composites

**DOI:** 10.3390/ma14185166

**Published:** 2021-09-09

**Authors:** Chao Feng, Jiaxing Huang, Peihui Yan, Fei Wan, Yunfei Zhu, Hao Cheng

**Affiliations:** School of Civil Engineering, Qingdao University of Technology, Qingdao 266033, China; huangjiaxing_123@126.com (J.H.); yph2113155@163.com (P.Y.); shuoyiecool@sina.com (F.W.); zyf13165046056@126.com (Y.Z.); 15866623753@163.com (H.C.)

**Keywords:** polypyrrole, hydrophilicity, cement, conductivity

## Abstract

The electrical properties of cement are gaining importance for the application in building construction. Polypyrrole (PPy) has been widely applied in most fields because of its excellent conductivity performance, environmental friendliness, easy fabrication, and other specialties. These features made them useful for self-sensing applications. In this work, waterborne polypyrrole (WPPy) was prepared via the chemical oxidative polymerization with three kinds of hydrophilic agents: sodium lignosulfonate (LGS), sodium dodecyl sulfonate (SDS), and sodium dodecyl sulfate (SLS), and then WPPy/cement composites were prepared by mixing cement with it. The contact angle, conductivity, and microstructure of WPPy were characterized by contact angle tester, four-point probes, and SEM. The composition, microstructure, and properties of WPPy/cement composites were characterized by FTIR, TGA, XRD, and SEM. The content of LGS was 40 wt%, WPPy got the optimal comprehensive performance, the conductivity was 15.06 times of the control sample and the contact angle was reduced by 69.95%. SEM analysis showed that hydrophilic agent content had great effect on the particle size of WPPy, the average diameter of WPPy particles decreased from 200 nm to 50 nm with the increase of LGS content. The results also showed that the adding of WPPy in WPPy/cement composites can significantly improve the conductivity and compactness, optimize the microstructure of cement composite. When the content of WPPy was 1.25 wt%, WPPy/cement composite showed the lowest resistivity and saturated water content of cement composite was 8 wt%. In addition, it could also inhibit the formation of Ca(OH)_2_ in the early hydration process.

## 1. Introduction

Concrete is one of the most widely used construction materials in the world [1,2,3], which has been developed for more than one hundred years. With its low cost, simple process, excellent strength, and good compatibility, it is widely popular in the civil engineering field [4,5,6,7]. In long service life, it is easily affected by harsh natural environment, fatigue damage, and other factors, resulting in some pernicious consequence such as deterioration and damage of internal structural materials, performance degradation, and some accidents [8,9,10,11]. The real-time monitoring and evaluation of cement-based structure using self-sensing technology has attracted more and more attention [12,13]. The traditional concrete-monitoring method is mainly realized by installing sensors on its surface or inside, which can be divided into strain gauge sensor and optical fiber sensor. The strain gauge sensor is mainly perceived by sticking strain gauge, which has the advantages of low cost and simple operation. Backer et al. [14] tested the strain gauge sensor in practical project, and conducted nondestructive testing on concrete beams. The result showed that the strain gauge had sufficient measurement accuracy in the construction process, which confirmed that the strain gauge sensor was feasible in practical project. In addition, optical fiber sensor is also a good sensor choice. Optical fiber sensor can monitor the micro cracks in cement-based structure through the subtle changes of the intensity, wavelength, frequency, phase, and polarization of the received optical signal [15]. Filho et al. [16] used optical fiber sensor to detect the improvement of concrete autogenous shrinkage crack by several super absorbent polymer-curing agents, and compared the sensing performance of mechanical strain gauge and optical fiber sensor, the result showed that the optical fiber sensor has higher detection sensitivity for microcracks. However, these sensors need to be embedded in the cement structure, which will affect the original structure of cement-based materials to a certain extent, and result in the reduction of the mechanical strength of the building structure. In order to resolve these problems, scholars proposed the concept of intelligent concrete materials, that can work as sensors. At present, research on self-sensing cement-based materials mainly focuses on piezoresistive cement sensors.

The resistivity changes when the cement component is subjected to tensile and compressive stress. Therefore, the strain or stress changes of cement structure can be monitored and sensed by simply measuring the resistivity of the cement structure. The intelligent conductive components mainly include metals [17,18,19] (such as steel fiber, steel slag, nickel powder) and conductive inorganic materials [20,21,22,23,24] (such as carbon fiber, carbon nanotubes, graphene, carbon black); however, there are fewer studies on conductive polymer. Polypyrrole (PPy) has many advantages such as simple synthesis process, environmental friendliness, and high conductivity as one of the conductive polymers. However, the PPy in intrinsic state cannot be dissolved in water and most common organic solvents, which also limits its processing performance. The main reasons are heavy interaction between the main chains, and the strong rigidity of the main chains. Over the years, many solutions have been put forward to solve these problems [25,26,27,28]. Zhang et al. [27] synthesized a pyrrole derivative (2-methyl-1,3-di(1h-pyrrol-1-yl) propan-1-one) with two N-substituted pyrrole rings by chemical polymerization. It has good solubility in many organic solvents and also shows good fluorescence property. Mondal et al. [29] synthesized water-soluble PPy microspheres with controllable size, monomer unit, and conductivity by adding sodium tetradecyl sulfate as surfactant, NaNO_2_ and HNO_3_ as oxidants. According to the concentration of NaNO_2_, the size of PPy microspheres ranged from 45 nm to 350 nm. Lv et al. [30] synthesized a new oligomeric PPy derivative: oligomer (N-(methacryl 12-dodecanol) pyrrole) by the method of side chain-induced solubilization. It shows excellent solubility, film-forming ability, and thermal stability in chloroform, tetrahydrofuran, and other organic solvents. But the conductivity is only 3.5 × 10^−5^ s/cm at room temperature.

In this paper, the hydrophilic modification of PPy with several hydrophilic agents was carried out by chemical oxidation method to prepare waterborne polypyrrole (WPPy). Then WPPy/cement composites were prepared by compounding WPPy with cement. We studied the effects of types and contents of hydrophilic agents on the properties and microstructure of WPPy. Moreover, the effects of WPPy on the microstructure, hydration process, and electrical properties of WPPy/cement composites were studied.

## 2. Experimental Section

### 2.1. Materials

The pyrrole monomer (Py) was purchased from Sinopharm Chemical Reagent Co. LTD. (Beijing, China). Iron(III) chloride hexahydrate was obtained from Tianjin Beichen Founder Chemical Reagent Co., LTD. (Tianjin, China). Three hydrophilic agents which include sodium lignosulfonate, sodium dodecyl sulfonate, and sodium dodecyl sulfate were supplied by Tianjin Guangfu Fine Chemical Research Institute (Tianjin, China). Anhydrous ethanol (C_2_H_6_O) was bought from Tianjin Beilian Fine Chemicals Development Co., LTD. (Tianjin, China). N_2_ (99.999% purity) was purchased from Qingdao Ludong Gas Co., LTD. (Qingdao, China). All solutions were prepared with deionized water. The preparation process of WPPy cement-based composites was as follows. Portland cement (P.O 42.5) was bought from Qingdao fangyuyuan building materials Co., Ltd. (Qingdao, China). Water reducing agent (PCA-I Polycarboxylic acid superplasticizer) was supplied by Jiangsu Subote New Materials Co., LTD. (Nanjing, China). All of the chemical reagents were of analytical grade.

### 2.2. Preparation of Hydrophilic PPy

The hydrophilic PPy products were prepared as follows. First, hydrophilic agent (sodium lignosulfonate (LGS), sodium dodecyl sulfonate (SDS), and sodium dodecyl sulfate (SLS)) was dissolved in 100 mL deionized water to prepare the solution. After the hydrophilic agent is dissolved, 40 mL ethanol was added to make a better dispersion of Py, followed by the addition of distilled Py (2 mL, 28.8 mmol). After stirring for 5 min, an aqueous solution of iron (III) chloride hexahydrate (62.5 mL, 0.115 mol/L) was added dropwise to initiate polymerization at a rate of 3 s/drop by a dropping funnel. Second, the flask was filled with nitrogen for 30 s to keep the N_2_ atmosphere and the solution was stirred for 12 h under ice water bath. Then, the products were collected by vacuum filtration. The obtained products were rinsed by alcohol and distilled water to remove excess reagents and other byproducts. Finally, the WPPy products were dried at 60 °C. WPPy samples obtained using different types of hydrophilic agents such as LGS, SDS, and SLS (marked as PPy/Ai, PPy/Bi, and PPy/Ci) were coded, as shown in Table 1. The experimental protocol employed to synthesize WPPy is shown in Figure 1 step 1.

### 2.3. Preparation of WPPy Cement-Based Composites

The water cement ratio was 0.4 by the consistency test of cement paste [31]. The dimensions of the specimens were 20 mm × 20 mm × 20 mm. The fabrication process is summarized as follows (Figure 1 step 2). First, WPPy was dissolved in a centrifuge tube containing water, and then spread out on ultrasound for 3 min. Second, the dispersed WPPy and water-reducing agent were added into the cement and stirred at high speed for 240 s. Third, the 20-mm test block cube with inserting copper sheet as electrode was prepared. Finally, the cement sample was put into water for curing and then taken out for testing after 28 days [32]. The specimens were dried in oven at 60 °C for 3–24 h several times until constant weight, weighting and testing conductivity after drying. The water content X was calculated by Equation (1), where M_0_ stands for the weight of the specimens being dry, Mi is the weight of the specimen after drying.
(1)$$X=\frac{M_{i}-M_{0}}{M_{0}}$$

### 2.4. Analysis and Characterization

The electrical conductivities of the WPPy were measured by Four-Point Probe (ST2253, Suzhou Jingge Electronic Co., LTD., Suzhou, China) employing the method on a pressed pellet according to Formulas (2) and (3) (Figure 2a). The pellets with 10-mm diameter and 1-mm thickness were obtained by subjecting the powder samples to a pressure of 30 Mpa.
(2)$$\rho =G\left(\frac{V}{I}\right)G\left(\frac{W}{S}\right)D\left(\frac{d}{S}\right)=\rho_{0}G\left(\frac{W}{S}\right)D\left(\frac{d}{S}\right)$$
(3)$$\sigma =\frac{1}{\rho }$$
where ρ is the resistivity; ρ_0_ is the resistivity-resistivity measurements, σ is the conductivity, G(w/s) is the sample thickness correction function, D(d/s) is the correction function of sample shape and measurement position. G(w/s) and D(d/s) could be found in the manual.

The contact angle was provided by Contact Angle Tester (SDC-200, Dongguan Shengding Precision Instrument Co., LTD., Dongguan, China). Thermogravimetric results were obtained by TA Instrument thermogravimetric analyzer (TGA, TAQ600, Inno Instrument Co., LTD., Weihai, China) at a heating rate of 10 °C/min from 25–800 °C under N_2_ atmosphere. Field emission scanning electron microscope (SEM, JSM-7500 F, JEOL Co., LTD., Tokyo, Japan) was used to observe the micromorphology of the samples. The Fourier transform infrared spectra (FTIR, Nicolet 6700, ThermoFisher Co., LTD., MA, USA) analyses were carried out with the KBr pellet method. The resistivity of WPPy/cement composite was measured by the voltmeter-ammeter method, Figure 2b.

## 3. Results and Discussion

### 3.1. FTIR Spectroscopy

Figure 3 shows the FTIR spectra of the intrinsic PPy and WPPy in the region of 4000 to 500 cm^−1^. The spectra showed a rich band fingerprint region. The peaks at near 1500 cm^−1^ and 1450 cm^−1^ could be attributed to stretching vibration of C−N and C−C on the pyrrole ring. The absorption peak at 1550 cm^−1^ corresponds to the C=C stretching vibration bond of the pyrrole ring [33]. These characteristic peaks were consistent with the characteristic peaks of WPPy. The broad band at 1300 cm^−1^ demonstrates the C−H and N−H in-plane deformation vibration. The peak at about 3400 cm^−1^ and 1600 cm^−1^ could be attributed to the stretching vibration peak of O−H. The characteristic absorption peak of WPPy can be found in other three WPPy spectra, which also confirmed the successful doping of dopants in WPPy chains. The strong peak near 1160 cm^−1^ and 890 cm^−1^ presents the doping state of WPPy. Additionally, the characteristic peaks near 1030 cm^−1^ and 1550 cm^−1^ are caused by the tensile vibration of S=O in the hydrophilic molecule [34], while the absorption peaks near 1450 cm^−1^ was the characteristic absorption peak of −OCH_3_ in the hydrophilic molecule. It could be seen that the S=O absorption peak intensity of PPy/C4 was significantly higher than the others, which corroborated WPPy doped by LGS has optimal hydrophilic properties. The absorption peak from 960 cm^−1^ to 1000 cm^−1^ corresponds the calcium silicate hydrate C_3_S_2_H_3_. There is an extremely strong and narrow absorption peak near 3640 cm^−1^, which is the hydroxyl group vibration absorption peak of hydroxyl calcium hydroxide Ca(OH)_2_, and the absorption peak at 1100 cm^−1^ and 3600 cm^−1^ was the characteristic peak of ettringite AFt.

### 3.2. SEM

From the Figure 4, it was obviously that there was a variation rule of particle size with the change of the content of hydrophilic agents. With the change of doping amount of LGS, the average diameter of prepared WPPy particles decreased from 200 nm (Figure 4b) to 110 nm (Figure 4c) and 50 nm (Figure 4d). The results showed that the amount of hydrophilic agent significantly affected the formation of WPPy particles, and the slightly higher amount of hydrophilic agent was beneficial for the nanometer of WPPy particles. Excessive LGS could reduce the polymerization rate of Py, resulting in slower oxidation of Py monomers and intermediates, slower chain growth of the WPPy, lower molecular size of the product and smaller particle size of the WPPy particles. When doping content was low, the transfer of electron transfer between the hydrophilic agent and WPPy changed the energy band structure of WPPy. In a high stage, after meeting the limit doping, too many hydrophilic agent molecules were connected with WPPy through hydrogen bond, providing hydrophilicity and dispersion. It was shown that LGS not only acted as a doping agent, but also acted as a dispersion agent.

Figure 5 showed the SEM images of cement paste with different curing ages. The hydration reaction began at the curing age of 1 day (Figure 5a). C−S−H gel formed a stress−supporting viscoelastic skeleton solid. Most hydrated products were thin and netted AFt, and there were a few flaky Ca(OH)_2_. The cement paste was not dense and porous, and had certain mechanical strength. After 3 days of curing (Figure 5b), it could be observed that the hydration products were irregular shaped AFt, some were small pieces, some were long strips, which were bigger than the samples in 1 day. The long strip AFt and C−S−H gel were connected to the space grid structure, which provided early cement strength, but there were still some gaps in the cement. At 7 days curing (Figure 5c), it could be observed that there were lump of AFt, and the cement strength further increased. After 28 days of curing (Figure 5d), the hydration reaction of cement was basically completed, showing the porous structure of cement, and the cement strength was the highest.

Figure 6 showed the SEM images of the WPPy/cement composites with different curing ages. After 1 day curing (Figure 6a), WPPy coated on the surface of the rod AFt, and C−S−H gel had just formed, there were still some pores inside. After 3 days curing (Figure 6b), it could be seen that the cement compactness had improved compared to 1 day (Figure 6a). WPPy significantly improved the cement compactness, and optimized the microstructure of cement. WPPy and AFt overlapped each other, the rod AFt became thicker, and the strength was further improved. After 7 days curing (Figure 6c), the hydration products were mainly bonded long rod-shaped and block shaped AFt, and WPPy was also attached to them. Compared with the cement composites of 3 days (Figure 6b), the morphology changed little, but the density of cement stone improved. After 28 days curing (Figure 6d), the hydration reaction of cement was basically over, and the small spherical WPPy/cement dispersed on the surface and voids of lumped AFt and CaCO_3_.

Comparing the SEM images of cement paste and WPPy/cement composite at the same curing age, it could be found that the hydration process of WPPy/cement composite was slower than the cement paste, and the hydration reaction rate of cement was reduced after combining WPPy and cement. After 1 day curing, the C−S−H gel in cement paste was lapped into the network, while the hydration products in WPPy/cement composites were only random-lapped short rod AFt. The morphology of WPPy/cement composites changed little after 3 days and 7 days curing, and the hydration products were mainly short rod-shaped AFt. The morphology of hydration products AFt changed greatly from 3 days to 7 days, the rod AFt turned into lump which enhanced the early strength. At curing age of 28 days, the cement paste had a dense structure, flat surface, and high strength. Some hydration products in WPPy/cement composites were not lumped due to the influence of WPPy.

### 3.3. TGA

Figure 7 showed the TGA and DTG curves of cement paste and WPPy/cement composite samples with different hydration ages. It was obvious that the TGA curves of the two materials were different in the four ages. In hydration of cement, the cement hydration reaction forms hydrated products such as C−S−H gel, Ca (OH)_2_, CaCO_3_, and ettringite. Generally, for the TGA curves of cement and cement-based composites, there were three obvious weight-loss regions. The first region was in the temperature range from 70 to 140 °C, corresponding to the loss of bound water in some hydration products, such as C−S−H gel and ettringite. The second region occurred between 350 and 450 °C, because of the mass loss caused by dehydration and decomposition of Ca(OH)_2_. The last weight loss field showed thermal decomposition of CaCO_3_ at high temperature (range from 650 to 800 °C).

It was obvious that for the 1 day curing age cement paste, there were rapid quality decline areas at 350–450 °C and 650–800 °C. However, WPPy/cement composite at this age does not show a peak of rapid mass reduction at 350–450 °C. It could be speculated that WPPy absorbed part of water during the hydration process of cement, which affected this process. Compared with the TGA curve of 3 days curing age, it could be speculated that WPPy inhibited the formation of Ca(OH)_2_, mainly because of the adsorption of water molecules by WPPy particles. Moreover, the residual mass of WPPy/cement composite was 5.00% higher than that of cement paste, which indicated that there was still part of cement that did not participate in the hydration reaction and cannot be decomposed by heating. The residual mass of WPPy/cement composite of 3 and 7 days curing age was less than cement paste, which resulted by the poor thermal stability of WPPy. It can be observed that the heat loss peak on the thermogravimetric curve was basically the same as cement paste. The compositions of hydration products of these two kinds of cement were basically the same, but the contents were different. The TGA curves of 28 days curing age of cement paste was similar to that of WPPy/cement composite; the mass reductions were 1.85% and 1.64% at 350–450 °C, 7.66% and 5.92% at 650–800 °C. The doping of WPPy could slow down the hydration process of cement, and inhibit the formation of Ca(OH)_2_ in the early hydration process.

### 3.4. Hydrophilicity of WPPy

Figure 8 showed the contact angle of WPPy. According to the test results, with the increasing of hydrophilic agent addition, the contact angles of the three kinds of WPPy were decreased, which means their hydrophilicity was enhanced. It was proved that hydrophilic agents were effective to enhance the hydrophilicity of WPPy. The contact angle of PPy/C has hardly changed, reached the minimum angle at PPy/C6 which decreased 14.51% only. However, PPy/B had an obvious downward tendency, PPy/B6 decreased 63.10% compared to the control. It could be the different hydrophilic enhancement effects between sulfonic acid groups and sulfate groups. PPy/A had the optimum hydrophilic enhancement effect. With the increasing of hydrophilic agents content, the contact angle of PPy/A decreased first and then flattened, the contact angle of PPy/A4 was 40.54°, and it decreased 69.95% compared to the intrinsic PPy. With the LGS content increasing from 0 to 30 wt%, the contact angle of PPy/A kept decreasing. The reason is that the sample was in the first stage of recombination, the doping amount of LGS had not reached the threshold, and with the gradually increasing of sulfonic acid groups bound on the WPPy molecular chain, the hydrophilicity increased dramatically. When the content of LGS was more than the effective doping amount, the quality proportion of lignin in sample PPy/A increased, and its hydrophilicity was mainly from the LGS. Therefore, with the continuous increase of LGS content, the hydrophilicity basically does not change. Yang et al. [35] viewed that LGS was inserted among PPy molecular chains as dopants at low doping amount, and mainly attracts each other through positive and negative charges. At high doping amount, excessive LGS molecules are connected with PPy by hydrogen bond, which reduces the electron transmission rate, which further proved the experimental results.

### 3.5. Electrical Conductivity

The electrical conductivity of WPPy was plotted in Figure 9. It was clear that the conductivity increased first and then decreased with the increase of PPy/A, PPy/B, and PPy/C content, which was due to the fact that it had a threshold value of the hydrophilic agents as dopant in the polymerization. In the increase phase, the hydrophilic molecules were intercalated into WPPy chains, which changed the electronic occupancy of WPPy molecular orbital. The resistance of electron movement decreased which made the conductivity of linear conjugation WPPy to increase. It has been reported that the doping amount of small molecular sulfonic acid in the WPPy prepared by chemical oxidation polymerization method was about 25–30 mol% [36], which was also known as the effective doping level. In the second phase, the hydrophilic agent in the reaction reached the threshold value. Hydrophilic agents could also act as template, which brought the hydrophilic molecules into the resulting samples by its adsorption interaction with the polymer. Surfactant anions with bulky aliphatic chains could be steric barriers for the charge transport in WPPy chains, which resulted in two contrary effects on the conductivity of WPPy samples. The nucleophilicity of ions could affect the positive charge distribution of the WPPy main chain, affecting the charge transmission in WPPy. Therefore, the inserting of SLS between WPPy chains only slightly changed the molecular orbital electron occupation. It was reported that [37] the conductivity of sodium dodecyl benzene sulfonate−doped WPPy was only slightly increased, which was consistent with the experimental results. It could be defined that the content of PPy/A4 was the most suitable content, in which the conductivity was 15.06 times of the control sample and the contact angle was reduced by 69.95%.

Figure 10 showed the resistivity of WPPy/cement composites in dry condition. It was observed that resistivity of sample without WPPy was 9150 KΩ.cm. The resistivity of the WPPy/cement composite decreased with the increase of the WPPy content, while the conductivity increased. However, 0.25 wt% WPPy doping could effectively enhance the cement conductivity and, and the resistivity was 1470 KΩ.cm, which was reduced by 83.9% compared to the control group. The reason was that WPPy had high conductivity, and many tiny conductive units were formed in the cement-based materials after dispersing, which provided higher conductivity. With the increasing of WPPy contents, the resistivity decreased further. When WPPy content was 1.25 wt%, the resistivity reached the lowest value of 402 KΩ.cm, which was 95.6% lower than the control group. The reason was that the number of conductive units in WPPy/cement composite materials increased with conductive WPPy content, and then formed conductive networks. Factors affecting the conductive performance of cement−based composite materials are the resistance of composite dopants and the contact resistance of cement between them. With the increasing of WPPy content, part of the contact resistance between cement was replaced by WPPy, which decreased the resistance of composite cement materials and improved their conductive performance. With further increase of WPPy, the resistivity began to rise, and the reason was that the WPPy contained a lot of hydrophilic groups, a large amount of water was absorbed in the hydration reaction process, as a result, the hydration of cement was not completed, which caused structural defects in the materials.

Figure 11 showed the effect of water content on electrical resistivity of WPPy/cement composites. It was obviously that with increase of water content, the resistivity of all cement materials decreased significantly, which indicated that water content had great impact on the conductivity of cement-based materials. The resistivity of control group with 1 wt% water content was 818 KΩ.cm which was 91.1% lower than the dry control group. Resistivity of the other WPPy/cement composite materials also decreased to less than 100 KΩ.cm, and with the increase of water content, resistivity of each sample trended to decrease. The reason was that water molecules in the porosities of cement material had formed the new conductive paths. With the further increase of water content, the resistivity of samples with more than 8 wt% water content remained constant, indicating that the further infiltration of water molecules had little influence on the conductivity of cement materials.

### 3.6. XRD

Figure 12 showed the XRD patterns of WPPy, cement, and WPPy/cement composite. A strong diffraction peak can be seen clearly in 26.571° [38,39], which was attributed to silicon dioxide (011) crystal plane diffraction peak. In addition, there had an obvious diffraction peak at 29.409°, which is calcium carbonate (104) crystal plane diffraction peak. There was a more strong diffraction peak at 34.028°, which is the calcium hydroxide (011) crystal plane diffraction peak. The diffraction peak at 50.732° was the characteristic peak of C_3_S, and the characteristic peak of C_2_S is at 36.023°, which was also the characteristic peak of cement.

## 4. Conclusions

In recent decades, with the rapid development of smart building materials, conductivity cement composites have begun to appear in our vision, which make great contribution in several fields, such as nondestructive testing, intelligent sensing, and so on. In this paper, WPPy was prepared by doping hydrophilic agent, and the effects of the hydrophilic agent types and content were studied. Further, the WPPy/cement composites were prepared, and the effects of WPPy on hydration process and electrical property of cement were studied. The results showed that the conductivity and hydrophilicity of WPPy could be significantly improved by doping hydrophilic agent. The prepared WPPy with 40 wt% LGS doping showed the best properties, the conductivity was 15.06 times of the control sample and the contact angle was reduced by 69.95%. The SEM images showed that hydrophilic agent content had great effect on the particle size of WPPy, the average diameter of WPPy particles decreased from 200 nm to 50 nm with the increase of LGS content from 10 wt% to 60 wt%.

The doping of WPPy enhanced the conductivity of the cement-based composites, the resistivity had the minimum value of 402 KΩ.cm in dry state, which was reduced by 95.61% compared to cement paste. SEM images and TGA showed that WPPy doping could slow down the hydration process of cement, and inhibit the formation of Ca(OH)_2_ in the early hydration process. The hydration process of WPPy/cement composite in the same curing age was obviously slower than cement paste, it may be the absorption of water by WPPy which affected the hydration reaction process. In addition, this study provides reference for further research and the application of conductive polymer in the area of self-sensing cement.

## Figures and Tables

**Figure 1 materials-14-05166-f001:**
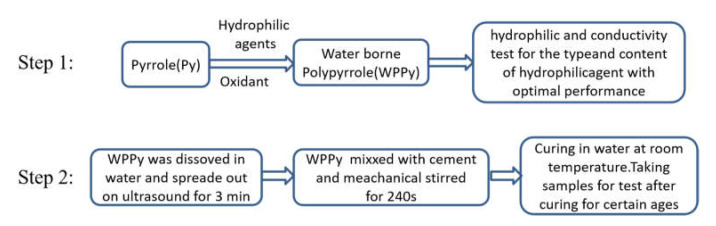
Experimental protocol employed to synthesize WPPy and WPPy/cement composites.

**Figure 2 materials-14-05166-f002:**
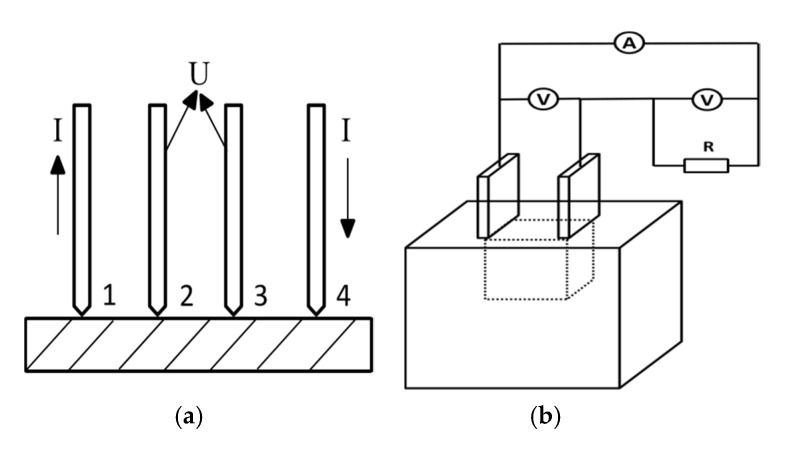
Test method for electrical conductivity: Four probe method (**a**), Voltmeter-ammeter method (**b**).

**Figure 3 materials-14-05166-f003:**
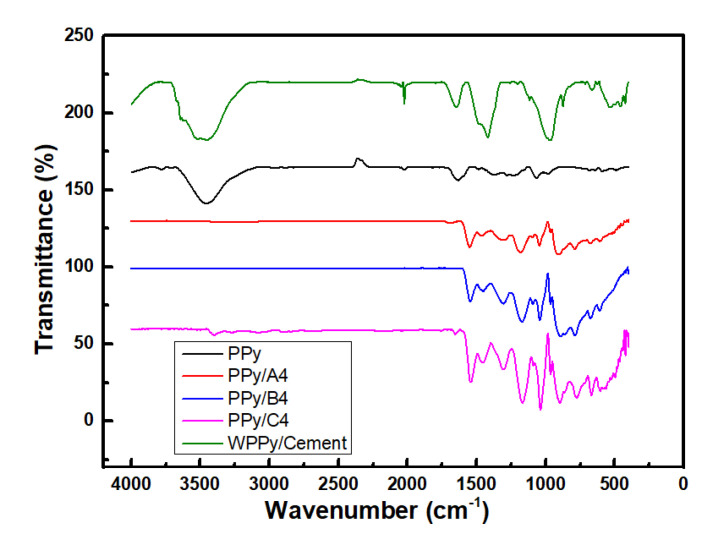
FTIR spectra of PPy, PPy/A4, PPy/B4, PPy/C4, and WPPy/cement composite.

**Figure 4 materials-14-05166-f004:**
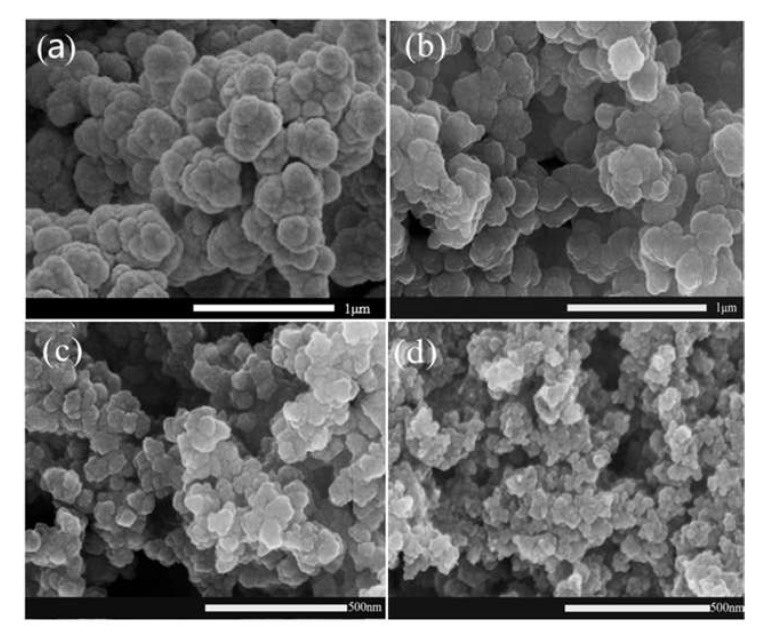
SEM images of WPPy doped with LGS, PPy (**a**), PPy/A1 (**b**), PPy/A4 (**c**), PPy/A6 (**d**).

**Figure 5 materials-14-05166-f005:**
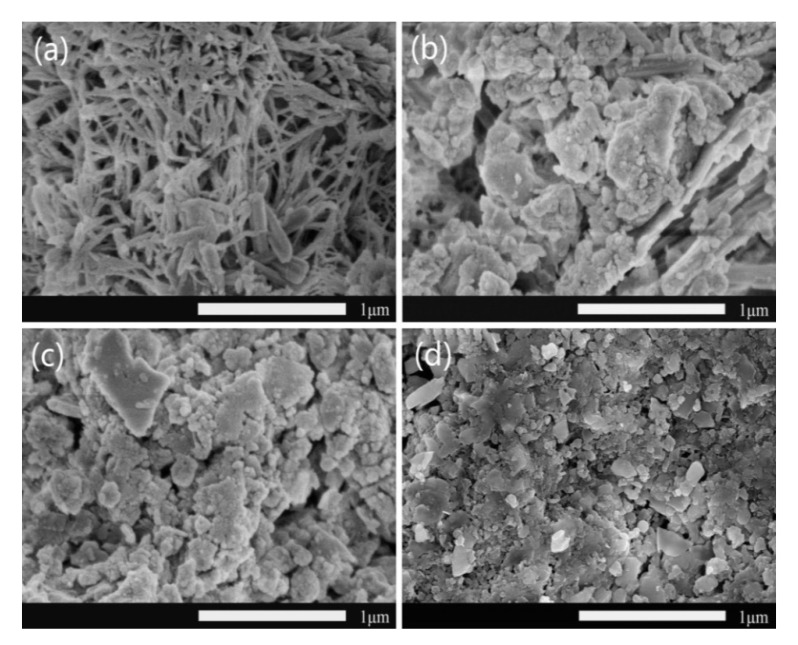
SEM images of cement paste with different curing ages. 1 day (**a**), 3 days (**b**), 7 days (**c**), 28 days (**d**).

**Figure 6 materials-14-05166-f006:**
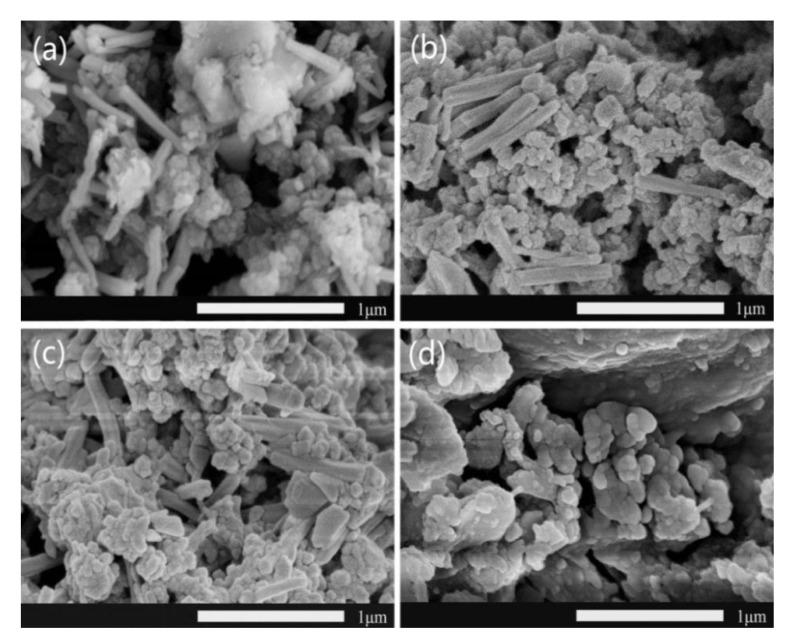
SEM images of the WPPy/cement composites with different curing ages. 1 day (**a**), 3 days (**b**), 7 days (**c**), 28 days (**d**).

**Figure 7 materials-14-05166-f007:**
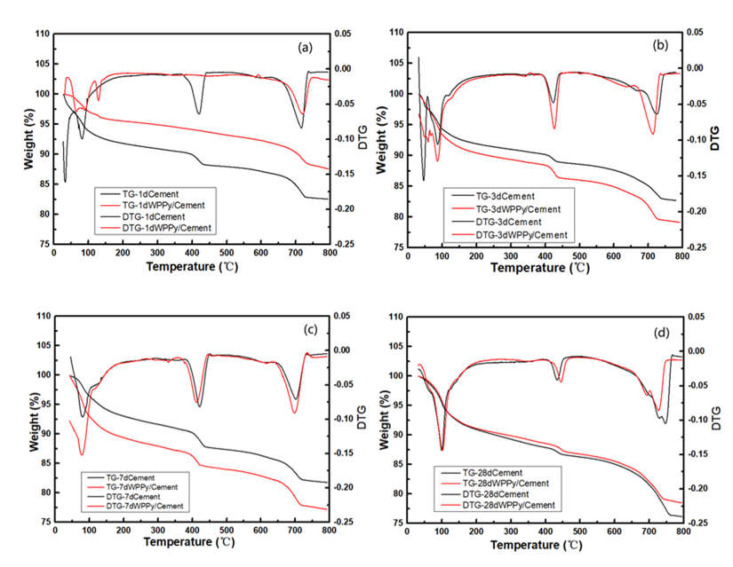
TGA and DTG curves of cement paste and WPPy/cement composites with different hydration ages. 1 day (**a**), 3 days (**b**), 7 days (**c**), 28 days (**d**).

**Figure 8 materials-14-05166-f008:**
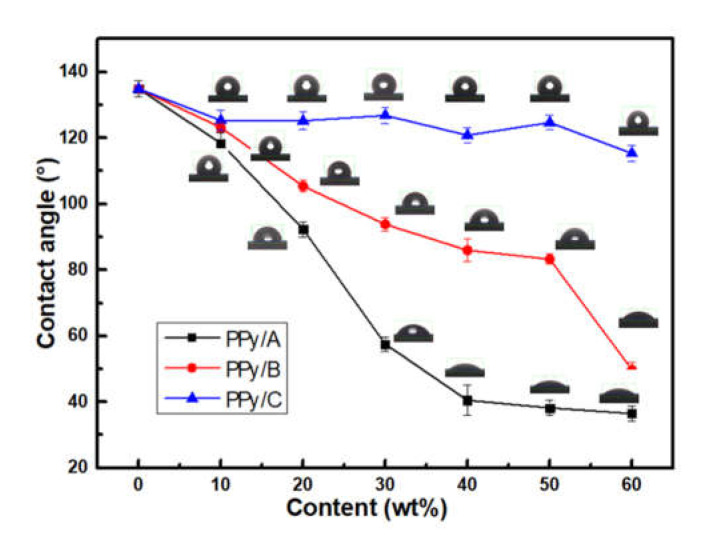
Contact angle of WPPy by different kinds of hydrophilic agents with different contents.

**Figure 9 materials-14-05166-f009:**
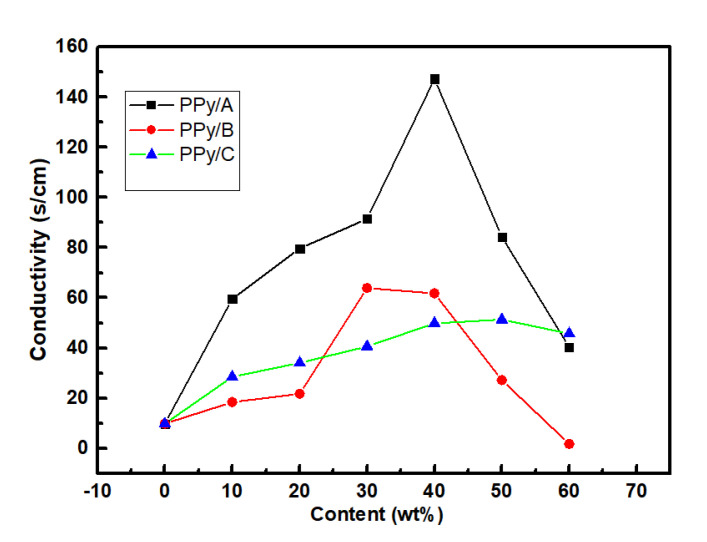
Conductivity of PPy/A, PPy/B and PPy/C.

**Figure 10 materials-14-05166-f010:**
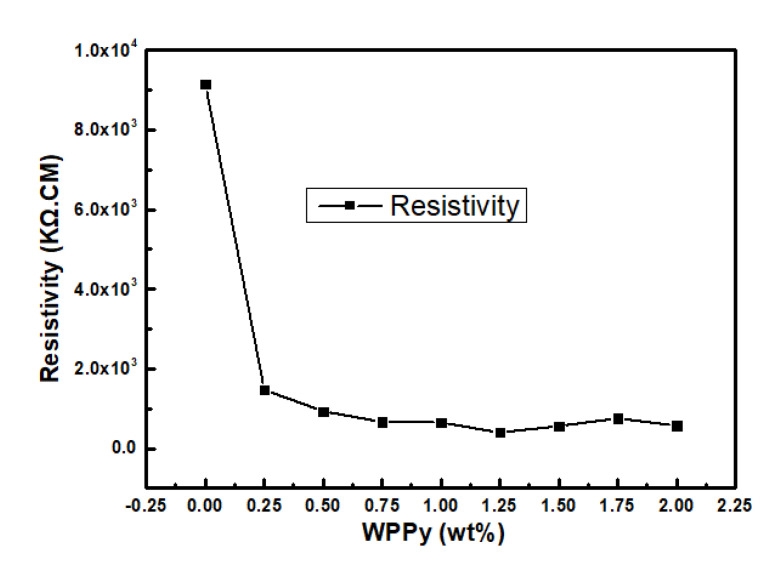
Resistivity of WPPy/cement composites in dry condition.

**Figure 11 materials-14-05166-f011:**
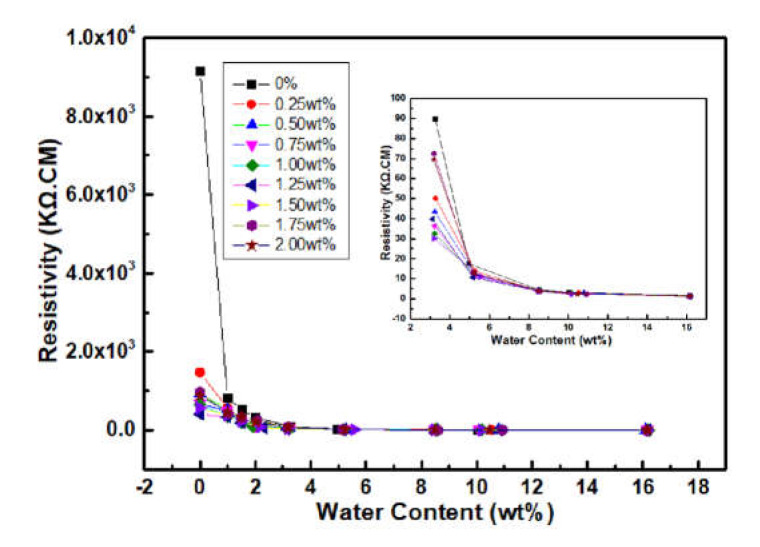
Resistivity of WPPy/cement composites with different water contents.

**Figure 12 materials-14-05166-f012:**
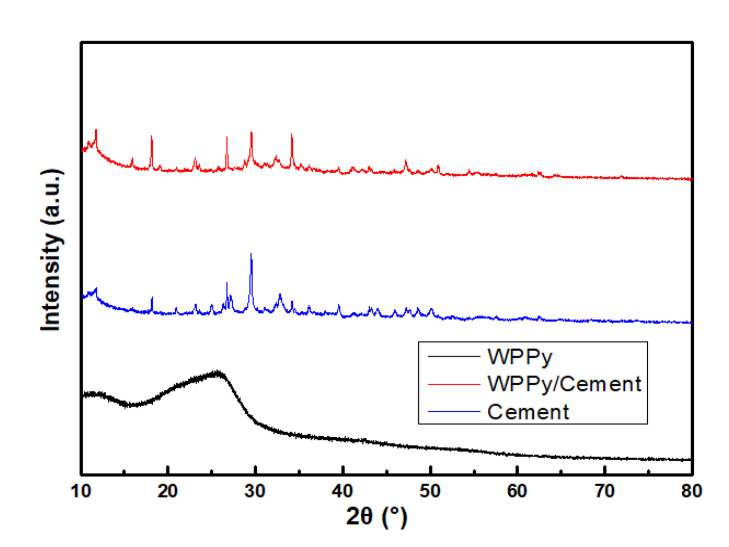
XRD patterns of WPPy, cement and WPPy/cement composite.

**Table 1 materials-14-05166-t001:** Polymerization formulas.

WPPy Samples			Py (mL)	Hydrophilic Agents (g)	Hydrophilic Agents (wt%)
LGS	SDS	SLS			
PPy/A1	PPy/B1	PPy/C1	2	0.2160	10
PPy/A2	PPy/B2	PPy/C2	2	0.4835	20
PPy/A3	PPy/B3	PPy/C3	2	0.8274	30
PPy/A4	PPy/B4	PPy/C4	2	1.2894	40
PPy/A5	PPy/B5	PPy/C5	2	1.9340	50
PPy/A6	PPy/B6	PPy/C6	2	2.9010	60

## Data Availability

The data presented in this study are available on request from the corresponding author.

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
