# Peer review of "Preparation and Properties of Waterborne Polypyrrole/Cement Composites"

_materials, 2021, doi:10.3390/ma14185166_

Round 1

Reviewer 1 Report

The abstract section is required for major revision which should include a short introduction, problem statement, methodology, and significant findings. Please avoid using abbreviations at the starting sentence, and spell out the full name before using it in the manuscript. This manuscript is hampered by several syntax errors. The methodology is appropriate but it could be more interesting if the flow chart can be provided.

The following suggestion and comments should be taken

  • What is WPPy?
  • Section 2.2. The material characterization details need to be added to the manuscript.
  • Table 1. The Hydrophilic agents are referring to all the three types of agents?
  • Based on Table 1, there are 6 different samples with 6 different hydrophilic agents. However, in Section 3.1, there is only 3 type of samples with 3 different hydrophilic agents has been analyzed, where are the others samples? The selection of samples for certain analyses needs to be clarified.
  • Table 1. Sample identification (PPy/A4, PPy B4, and PPy/C4) cannot be recognized.
  • Syntax error. Please check spacing in the relevant text. 
  • “S=O absorption peak intensity of PPy/C4 was significantly higher than the other.” Please extend the discussion on this sentence. 
  • Figure 2. Sample identification is not reflected in Table 1. Is any additional sample?
  • “Excessive sodium lignosulfonate could reduce the polymerization rate of Py,…” What is the limit value for the hydrophilic agent?
  • “…only acted as a doping agent, but also acted as a dispersion agent.” Please discuss the details of the role for each agent.
  •  Figure 4. What is the significance of SEM for cement paste that can be correlated with other images of cement composites? Without EDX analysis, how did the author confirms the existence of the mineral?
  • Section 3.3. Why there is the different curing ages has been selected for TGA analysis? 
  • At 70 C to 140C, the loss of water is usually related to water hydration. Please revise clearly.
  • “…with the continuous increase of sodium lignosulfonate content, the hydrophilicity does not change.” It is just refer to the quality portion of lignin in the sample? How did the author determine the quality of lignin towards hydrophilicity?
  • What did the author mean by hydrophilic agent C?
  • Section 3.6. What is the significance of the phase analysis towards the objective of the current study?

Author Response

Dear Reviewer:

Thank you very much for your suggestions and comments, this is the detail response to your comment.

Best regards!

Reviewer 2 Report

The paper "Preparation and properties of waterborne polypyrrole/cement cimposites" is interesting and within the scope of Applied Sciences. The use of smart materials in construction is a growing trend and any study about it is welcomed by the specialized scientific community. So In my opinion is a very interesting topic, but it must be improved in order to be published.

As the authors have not written the article in the format of the magazine, I cannot specify my comments line by line, I will make some common comments for the whole the paper.

The text should be revised, there are some errors, such as repeated words, semantic and grammatical errors. The placement of the citations of the authors should also be revised, in some cases they are not correct.

Abstract

 I would recommend revising the abstract. The meaning of all the acronyms used by the authorss should be included and to increase the quality of the article. Also it should be included why it is important to increase the conductivity of cement, since in my opinion this is the most important contribution of this article.

Experimental Section

The materials subsection should be reorganized. It would be good to make a table with the most important characteristics of the compounds to be used in the tests, in order to facilitate the reading for the rest of the researchers.

In all the studies carried out to study the properties of cements and concretes, the standards used for the preparation of the samples, as well as those governing the test to be performed, are indicated. It would be interesting if all these standards were included in the article so that other researchers could repeat the tests performed.

Discussion

The results obtained by the authors are well described in the text, but should be compared with results obtained by other authors in similar studies, to validate the findings described in the article.

Conclusions

In this section, it should be emphasized why it is important to increase the conductivity of cement pastes.

Author Response

Dear reviewer:

 Thank you very much for your comments and suggestions. This is the detail response to your comments.

Best regards!

Round 2

Reviewer 1 Report

  1. The authors made some revisions to the manuscript. As a suggestion, please show the specific changes (line/pages) that have been made in the revised manuscript. 
  2. Please add some knowledge contribution in the abstract.
  3. If can, please add some more critical discussion towards fundamental especially main significant results.
  4. I recommended this manuscript be accepted for publication.

Author Response

Dear reviewer:

 Thank you very much for your advice. The documents followed are my response and manuscript.
